# Influence of Stress and Emotions in the Learning Process: The Example of COVID-19 on University Students: A Narrative Review

**DOI:** 10.3390/healthcare11121787

**Published:** 2023-06-17

**Authors:** Alfredo Córdova, Alberto Caballero-García, Franchek Drobnic, Enrique Roche, David C. Noriega

**Affiliations:** 1Department de Bioquímica, Biología Molecular y Fisiología, Facultad de Ciencias de la Salud, GIR: “Ejercicio Físico y Envejecimiento”, Universidad de Valladolid, Campus Universitario “Los Pajaritos”, 42004 Soria, Spain; 2Department de Anatomía y Radiología, Facultad de Ciencias de la Salud, GIR: “Ejercicio Físico y Envejecimiento”, Universidad de Valladolid, Campus Universitario “Los Pajaritos”, 42004 Soria, Spain; alberto.caballero@uva.es; 3Medical Services Wolverhampton Wanderers FC, Wolverhampton WV3 9BF, UK; docdrobnic@gmail.com; 4CIBER Fisiopatología de la Obesidad y Nutrición (CIBEROBN), Instituto de Salud Carlos III (ISCIII), 28029 Madrid, Spain; eroche@umh.es; 5Department of Applied Biology-Nutrition, Institute of Bioengineering, University Miguel Hernández, 03202 Elche, Spain; 6Alicante Institute for Health and Biomedical Research (ISABIAL), 03010 Alicante, Spain; 7Department Cirugía, Oftalmología, Otorrinolaringología y Fisioterapia, Facultad de Medicina, 47003 Valladolid, Spain; dcnoriega1970@gmail.com; 8Departamento de Columna Vertebral, Hospital Clínico Universitario de Valladolid, 47003 Valladolid, Spain

**Keywords:** emotions, emotional intelligence, intellectual performance, learning, stress

## Abstract

Emotional instability and stress are the main disturbances that condition the learning process, affecting both teachers and students. The main objective of this review is to analyze the influence of stress and emotions (as part of stress) on the learning environment. Stress is a physiological mechanism that the organism develops to adapt and survive external and internal challenges. In this context, stress tends to be seen as a negative condition in the learning process when it is chronic. Extreme stress situations, such as the COVID-19 pandemic, can generate anxiety and frustration in students. However, other studies indicate that controlled stress can positively enhance the learning process. On the other hand, the quality and intensity of emotions resulting from stress can influence as well the learning process. Positive emotions are healthy and can promote optimal learning. Emotions lead to sentimental, cognitive, behavioral, and physiological changes, which will have a strong influence on intellectual performance. The activation of coping strategies constitutes a key mechanism for dealing positively with problems and challenges, generating positive emotions essential for the self-regulation of learning. In conclusion, correct management of emotions in stressful situations could promote effective learning through enhanced attention and capacity to solve problems.

## 1. Introduction

Learning involves the interaction of cognitive, emotional, and physiological elements. In this context, emotions are considered a very important part of the knowledge acquisition process. The Greek philosopher Plato (427–347 b.C.) already indicated that “all learning has an emotional basis”. Nevertheless, there are many definitions for emotions, and no definition is considered unanimous. In addition, stress is not a mere negative aspect; it is necessary for learning, becoming a positive aspect when it is under control [1,2,3,4].

Emotions occur when a person evaluates an event or situation as important [5]. For instance, the most recent stress situation (the COVID-19 pandemic) has caused an emotional disturbance in the general population. The field of education was particularly affected, increasing stress levels among students. This occurs because emotions are accompanied by sentimental, cognitive, behavioral, and physiological changes [6,7,8].

In 2020, the World Health Organization (WHO) declared pandemic status to COVID-19 infection due to the alarming levels of coronavirus spreading into the population [9]. The fears generated by exposure to COVID-19 resulted in anxiety and states of stress in all people and particularly among education stakeholders (teachers and students). Although direct measures were taken to safeguard the well-being of both teachers and students, the pandemic situation altered their emotional state, leading to burnout and decreased learning ability [10,11,12,13]. In other pandemics, such as the influenza pandemic, high levels of stress, anxiety, and low mood were observed in this case in healthcare workers [14,15]. These situations undoubtedly lead to increased fatigue and decreased intellectual performance. In the field of education, the pandemic situation has forced students to be at home, not attending universities and high schools, or alternatively attending in small groups. Online classes have certain advantages; some students, particularly those studying health sciences, require more resources making it difficult to receive suitable classes [16,17]. Otherwise said, students who usually follow their studies on a face-to-face basis were not very comfortable with the online experience and underwent a systemic shock. According to our personal experience as university professors, the key ways in which the pandemic has affected education and contributed to higher stress levels among students are:-Transition to remote learning: Universities around the world closed their doors to curb the spread of the virus. This situation forced teachers and students to adapt to online platforms for teaching and learning. This transition presented numerous challenges, including technical difficulties, limited access to resources, and a lack of face-to-face interaction with teachers and peers. We think that this could lead to feelings of isolation and disconnection.-Disrupted routines: Pandemic disrupted familiar routines and structures that students relied upon for learning. Students found difficulties in separating academic from personal lives. We suppose that this forced situation contributed to feelings of anxiety and stress.-Increased workload and academic pressure: Remote learning needed adjustments to teaching methods. Teachers had to redesign lesson structure, contents, and assessments for online delivery, resulting in an increased workload for students. In addition, the shift to virtual tests introduced new challenges, such as technological concerns about academic integrity. All this added additional pressure to students.-Limited social interaction: Isolation measures meant that students were unable to engage in face-to-face discussions, group projects, or extracurricular activities. All of them were important for their social and emotional well-being. We presume that the lack of social support exacerbated stress levels and feelings of loneliness.-Uncertainty about the future: This uncertainty included concerns about health, safety, and economic stability. In the case of students, uncertainty was focused on academic progression, graduation, and career projects. The constant changes in guidelines and regulations added more stress and anxiety levels as students had to adapt to ever-evolving circumstances.-Impact on mental health: The combination of the aforementioned factors, along with the general stress and anxiety caused by the pandemic, had a significant impact on the mental health of students. Actually, mental health has become a pressing concern in the education sector, requiring increased attention and support.

As university teachers, we have observed that the mentioned factors may contribute to students’ stress during the pandemic. This extreme situation has motivated us to write this narrative review. Nevertheless, it is important to note that the stress impact can vary among individuals and across different educational settings.

## 2. Objective

The main objective of this narrative review is to analyze the influence of stress and emotions (both components interconnected) in the learning process of university students. First, the COVID-19 pandemic will be taken as a recent example of learning disturbances. The review was mainly focused on academic stress, although there are other types of stress that will be briefly mentioned throughout the text, such as social/environmental, time, and future-related stress. Then, we wanted to know how emotions derived from stress influence the learning process. Since students experience stress in different ways, the review will focus on both positive emotions (resulting from good stress management) and negative emotions (resulting from poor stress management). The different aspects influenced by emotions in the learning process will be presented, including physiological and non-physiological components, the function of emotions, intelligence, and a brief mention of the impact of emotions on talented individuals. In this review, intelligence is considered as the ability to acquire and apply knowledge, think abstractly, reason logically, solve problems, adapt to new situations, and learn from experience. A key point of this narrative review is the information from a neurophysiological point of view in order to understand the role of emotions in the systemic response. The purpose of this information is to open interest in future pharmacological research or other lines in the emerging field of neuropsychology. The key points of this review and their connections are presented in Figure 1.

## 3. Methods

To conduct this narrative review, a comprehensive literature search was performed using the databases PubMed, Ovid MEDLINE, and EMBASE and the following search terms “EMOTIONS”, LEARNING”, “EMOTIONAL INTELLIGENCE”, “COVID-19”, and “STRESS”. Due to the huge amount of information, we selected some articles from the last ten years regarding “EMOTIONAL INTELLIGENCE” and “UNIVERSITY EDUCATION AND COVID-19”. The last combination was used to write Section 4. We combined the terms “LEARNING AND STRESS” and “LEARNING AND EMOTIONS” for Section 5 and Section 6. The search using the term “EMOTIONAL INTELLIGENCE” was used to write Section 6.3 and Section 6.4. Due to vast number of articles (more than 2500), we read (around 500 each author) carefully the titles to select articles related to university education and learning from the last 10 years. According to their particular experience as teachers at the university, each author proposed a selection of articles. Finally, we have selected those articles that communicate the data in a detailed way. It should be noted that many of them presented very similar messages and were excluded. Then, we read in more detail selected articles (around 30) and analyzed those important aspects (inclusion criteria) that could contribute to the development of the review: physiological and non-physiological components of emotions, the function of emotions, emotional intelligence, and the impact of emotions in talented individuals.

## 4. COVID-19, an Example of Learning Disturbances Due to Emotions and Stress

Emotions are manifested as a response to managing environmental changes and trying to maintain the subjects’ well-being [18]. The COVID-19 pandemic is the closest example that undoubtedly has generated fear and anxiety (negative emotions) in many segments of the population [12]. In the university context, students were susceptible to developing stress and depression, amplified by the isolating situation that the pandemic generated [19]. In this context, stress and anxiety difficulted learning and increased dropout. Therefore, students must balance individual stressors and education demands [19,20,21]. In general, stress tends to be seen as a negative condition at the beginning, although it is key for long-term adaptation [22,23]. This occurs because stress responses are mediated by the sympathetic-adrenal-medullary axis (SAM) (a neuroendocrine stress-response system), in particular by the hypothalamic-pituitary-adrenal (HPA) axis, culminating in the release of adrenal hormones such as cortisol. However, in the long term, hormones released are involved in the regulation of the immune system and inflammatory activities, contributing to such adaptation [2,23].

Teaching during a global pandemic was challenging [24,25,26,27]. However, before the COVID-19 outbreak, the mental health of young adults was already a global concern. Early reports during the COVID-19 pandemic showed that students did not prefer e-learning vs. face-to-face teaching [28]. However, in the last moments of the pandemic, the main part of studies reported positive perceptions of e-learning [29]. The change in this tendency occurred because learning-centered approaches were used to facilitate access to content. Despite all advantages, e-learning was not perceived positively in all academic contexts, particularly in health science education. In this line, the main drawback of this e-learning was the isolation and the impossibility of live practices necessary to acquire future professional competencies [30,31]. Most of the studied participants were worried about concentration capacity, academic progress, future plans, and academic performance [32]. Therefore, isolation played a key role in learning stress during the COVID-19 pandemic. However, students enrolled in campus wellness in the USA with no severe isolation were more confident and optimistic than their home-quarantined counterparts [33]. Contrary to this unhabitual situation during the pandemic, around 25–30% of students surveyed reported moderate to extremely severe anxiety, depression, and stress scores as a result of the fear of e-learning during home quarantine in Spain [34]. To support this information, isolation in quarantine seems to be an independent risk factor for adverse mental health outcomes, increasing 2.8-fold the risk of depression, 2.0-fold anxiety, and 2.7-fold stress response disorders [35,36].

## 5. Stress and the Learning Process

Evidence strongly assesses that stress has a key influence on the learning process. First of all, stress affects memory recall. It is well known to most students that the stress undergone when facing an evaluation test can make it difficult to recall information that might be available in less stressful circumstances [37]. Nevertheless, the impact of stress on the process of encoding information into memory is controversial. Some authors defend that stress impairs memory functions [38]; meanwhile, others show that stress enhances memory encoding [39]. However, it is difficult to determine the factors that are responsible for the reported discrepancies. It seems that stress could be considered a positive component when it is under control and a negative factor when the circumstances are clearly adverse (no possibility of control) [1]. Otherwise said, when stress is moderate, it can enhance learning and memory. However, when stress becomes excessive or chronic, it can have negative effects on learning. High levels of stress can impair cognitive function, attention, and memory retrieval, leading to difficulties in concentrating, processing information, and retaining new knowledge. Therefore, chronic stress negatively affects the physical and mental health of students and hinders academic performance [40,41]. During the first years at the university, students experience high levels of stress and anxiety, largely due to the new situation and the competition between students themselves [42,43]. Learning stressors are also associated with personal stressors, such as family demands, work, or sports, that compete with the demands of education. Undoubtedly, the pressure felt by the student exacerbates their state of stress and anxiety [43].

### Neurophysiology of Stress

Stress acts on different neural pathways and brain locations critical for memory. It has rapid effects by producing an increase in dopaminergic and noradrenergic activity in the prefrontal cortex [44,45]. It acts as well through the hypothalamus-amygdala axis (HPA) to regulate the secretion of stress hormones adrenaline and noradrenaline [46,47,48,49,50]. These hormones stimulate the vagus nerve and ultimately influence the hippocampus, amygdala, and prefrontal cortex, among other regions [47,48,49,50,51,52].

## 6. Emotions and the Learning Process

Emotions are interpretations of the internal and external environment. The perceived information is used subsequently for action [2,23]. In addition, emotions have a motivational function that predisposes them to repeat behaviors linked to positive feelings. In this line, emotion regulation is multidimensional and includes (a) awareness, understanding, and acceptance of emotions; (b) the ability to control impulsive behaviors during distress; (c) the ability to use appropriate situational strategies to modulate the intensity and duration of emotions; and (d) the willingness to experience negative affective states in order to engage in meaningful life activities [8]. Loss of control, stress, anxiety, and threat perception have been found to be involved in the reaction to emotions, resulting in increased anxiety and feelings of helplessness [8].

Regarding classification, emotions seem to follow a complex interpretation pattern depending on whether they are positive, negative, or neutral, pleasant and unpleasant, problematic, individual or collective [53,54]. Altogether, an adequate classification for emotions could be: (a) primary or innate, also called basic, pure, or elementary; (b) secondary or acquired, also known as social. In general, the primary would include fear, surprise, anger, rage, disgust, sadness, and happiness, among others. The secondary includes guilt, shame, contentment, jealousy, acceptance, resignation, and pleasure, among others.

However, in the learning process, classification using the terms “positive”, “negative”, and related terms seem to be more appropriate. Therefore, negative emotions affect well-being and provoke a desire to avoid or evade. These include anger, aversion, fear, anxiety, sadness, and shame, among others. Positive emotions are considered healthy because they positively affect the well-being of the individual. They favor the way people think, reason, and act, including joy, humor, love, and happiness, among others. To complete this classification, ambiguous emotions that do not fit with the previous criteria should be considered, including hope, surprise, and compassion. Finally, toxic emotions with a very negative component are considered in this classification, with envy being the most representative [55].

In addition, it is also necessary to differentiate between emotions and emotional states. Emotion is characterized by an alteration of the physiological response that predisposes one to an organized, systemic response [23,56,57]. However, emotional state refers to mood, which is of longer duration and lower intensity. Emotions are of short duration and are triggered by a specific stimulus [58]. Coming back to the example of the COVID-19 pandemic, the first evidence is that the emotional impact on each individual is very particular. For some, their emotional well-being decreased as they did not know how to cope with the uncertainty, anxiety, boredom, and even sadness that staying at home implied. For others, the same situation was a challenge turning these emotions into positive ones [59,60].

Therefore, it is well known that positive emotions support learning by increasing intellectual, physical, and social resources, favoring creative development, and improving coping strategies [61,62]. These emotions end up modifying the perception of learning. Therefore, the success achieved after the effort should be considered a non-measurable entity. Success is not an academic grade, a job, a salary, or a qualification from the environment. Success is mainly an emotional and individual state, which is not defined by outside references but from personal introspection. In this context, the concept of success fits with the definition stated by John Wooden in 1997 [63] “Success is a peace of mind that is a direct result of self-satisfaction in knowing you did your best to become the best that you are capable of becoming”.

### 6.1. Neurophysiology of Emotions

The emotional state is composed of an overt element characterized by physical sensations and another characterized by a concrete feeling (conscious process) [64]. Both emotional states and feelings are regulated in different anatomical structures. Concrete feelings are regulated by the cerebral cortex, partly the cingulate and the orbitofrontal cortex. Emotional states are regulated by joint action with nervous, endocrine, and skeletal-motor responses. The amygdala is the intercommunication structure of somatic expression of emotions (hypothalamus and brainstem nuclei) and is the system of interpretation of concrete feelings, especially fear (cingulate, parahippocampal, and frontal cortex) [65]. This occurs because the brain builds a permanent mental image during the emotional state with a mental map that characterizes the state of the body (viscera, musculoskeletal system) [64]. When the brain detects emotional stimuli, it sends specific commands to the CNS, NES (neuroendocrine system), ANS (autonomic nervous system), and the musculoskeletal system. This results in different reactions to emotions. The involuntary physiological response seems to be the first reaction to emotion. The psychological response comes after, including the way in which information is processed. The behavioral response generates a change in mood. Based on the systems involved in emotions, as well as the development of processes and functions, five components are identified [5]:(a)Physiological, which fulfills the function of regulation of organ systems, depending on the CNS, ANS, and NES [2,66,67]. These systems regulate physiological and emotional responses influencing the unconscious and instinctive behavior (important in survival). Many of these innate and primitive behaviors are altered by the brain cortex [68]. For instance, human unconscious behaviors, such as confidence, hope, joy, guilt, and despair, are influenced by conscious moral, social, and cultural codes [66,67].(b)The cognitive component is linked to information processing and has the function of evaluation.(c)The motivational component is linked to the CNS, which prepares and directs actions. In emotional states, there are more active CNS locations [68].(d)Motor expression fulfills a communicative function by informing the behavioral reaction and intention.(e)The subjective aspect serves to monitor the internal state of the organism and its interaction with the environment.

The fundamental function of the amygdala (subcortical structure in the internal temporal lobe) is the processing and storage organ of emotional reactions. It produces short-term adaptation, which allows an increase in rapid, unconscious responses that, although not very precise, are effective. The central nucleus coordinates efferent information that gives rise to both autonomic (sympathetic and parasympathetic), endocrine, and behavioral emotional responses [57,64,69]. The amygdala facilitates the formation of stimulus associations and helps to establish the emotional meaning of different situations [70].

Emotional information follows two pathways (one fast and one slow) to the amygdala. A third, equally important pathway (hippocampus-amygdala) underpins contextual conditioning. Through the thalamus-amygdala connections, an affective process with simple sensory features is elaborated. Through the thalamocortical connections, the complex process without affective components is produced. Through the cortico-amygdaline connections, the emotional component contributed to the complex information elaborated in the cortex [66,71].

The final process results in adaptive changes and reorganization of CNS due to its neuroplasticity. These changes occur both by learning and by adaptation to internal or external situations. In this process, activities such as neurogenesis might be considered. This results in the subjective, behavioral, and emotional components of emotions [62,72,73,74,75]. In this sense, and from the point of view of stress, the influence of personal traits, self-perception, and previous experiences must be considered. They depend to a large extent on the type of stressor [76]. This explains why coping requires behavioral and cognitive effort to manage stressful situations. In this context, some strategies are considered to focus on the problem (person-situation relationship) and others on the emotional disturbances resulting from the stressful situation [77,78].

Therefore, the left hemisphere of the brain, the rational part of the brain, is where structured functions locate, i.e., language. The right hemisphere, on the other hand, is the emotional hemisphere, which governs subjective feelings, i.e., the ability to appreciate art and music. The emotional part processes much faster than the rational mind. This last one establishes the cause-effect relationship. This is supported by objective evidence, leading to a re-evaluation of the situation. This can give the possibility to change a previous decision [68,69,70,71].

Regarding neurotransmitters, the four important pathways in an emotional-motivational sense are those mediated by dopamine, serotonin, noradrenaline, and endorphin. The release of dopamine translates motivation into action. Upon stress exposure, dopamine mediates by selecting the optimal response for coping with stressful situations [76]. Different hormones and neurotransmitters complement this reward emotional response, such as testosterone and oxytocin, respectively [76]. Finally, the neurotransmitter serotonin, known as the “mood hormone”, is related to well-being and helping to manage stress. In addition, serotonin inhibits anger and regulates temperature, mood, appetite, sleep, falling in love, among others [76]. The result is that emotions have three clear functions: adaptive, social and motivational [79,80] (Table 1).

### 6.2. Function of Emotions

All emotions, pleasant or unpleasant, are necessary and useful, as they allow one to quickly assess situations. They inform if something is important for well-being. Emotions are very important for increasing learning capacity and memory, playing an important role in decision-making [81].

Coming back to the example of COVID-19, all functions of emotions were altered at the current time of the pandemic. Studies on Argentinean populations showed a high level of uncertainty and fear in relation to COVID-19 [82]. Similar data were obtained in India, where they expressed a high degree of concern and uncertainty about COVID-19. A higher percentage of the population indicated the need for psychological support to reduce the impact on their mental health [83]. In this line, in Spain, the main part of students felt fear, anxiety, stress, and uncertainty related to COVID-19 infection [84]. Others have obtained similar results in studies on the return to clinical learning after the COVID-19 outbreak [85]. Depending on the vulnerability of the subject, and once the acute phase is over, it can be observed a functional or adaptive coping response or a dysfunctional, maladaptive, or counterproductive coping response [86,87,88].

Understanding the biological basis (Section 6.1) of executive functions may provide more insight into the mechanisms underlying the effects of stress on cognition [89,90]. One of the main functions of emotions is to facilitate the appearance of appropriate behaviors facing stress [81]. The expression of emotions allows us to predict the behavior associated with them. In this line, knowledge of an executive task can positively modulate performance [91,92]. Nevertheless, disturbances coming from the amygdala, such as fear, can negatively affect learning and memory storage [93,94,95]. Therefore, knowing how to modulate emotions in stressful situations could promote effective learning. Emotions can act on the psychological processes responsible for focusing attention or solving problems [96]. This emotional self-regulation, understood as the ability to motivate oneself, would be one of the dimensions encompassed by the so-called emotional intelligence [97].

### 6.3. Emotional Intelligence

Emotional intelligence (EI) contributes to an individual’s ability to adapt socially, work more effectively in teams, perform better, and cope more effectively with stress and other forms of environmental pressure [98,99]. Therefore, EI functions as an essential predictor of students’ learning and cognitive health [100,101]. For instance, students and faculty have managed the learning process during the pandemic through their emotional intelligence and cognitive engagement in blended learning environments [102]. Higher education institutions have managed the problem of campus closure by switching from face-to-face to online classes [103].

EI involves the perception, processing, regulation, and management of emotions. From a technical point of view applied to learning, EI is defined as a capacity-based skill that enables training in specific competencies that can be directly applied to a specialized field. Otherwise said, EI can be used to address specific aspects of the teacher-student relationship [103]. EI includes everything that is not covered by academic intelligence, such as impulse control, self-motivation, and social relationships, among others [54,104]. Therefore, EI links the thinking part of the brain (neocortex) with the emotional part and the limbic system [105,106].

The application of EI has been proposed in clinical, social, educational, and organizational settings [107]. Various studies [106,107,108,109] have explored the relationship between individual differences in EI and academic outcomes. According to their results, it appears that the construct of trait EI may serve as a moderator of the relationship between intelligence and school performance.

The abilities that constitute the EI construct are multiple. However, at present, two major models of EI can be found: (a) ability model, which studies abilities that deal with affective information [106,109]; and (b) mixed models that link emotional and cognition abilities with personality [105]. From the ability model, EI is expressed as a set of abilities to perceive access, understand, and regulate emotions to promote emotional and intellectual growth [109]. In this context, self-awareness and self-motivation (dimensions of EI) have a direct, positive, and significant impact on study habits [110]. However, other dimensions of EI (emotion regulation and social skills) have less influence. The structure of emotional awareness is based on cognitive schemes that are different among individuals. Emotional awareness undergoes a structural transformation following a hierarchical development defined by five levels (in progressive order): physical sensations, action tendencies, individual emotions, mixtures of emotions, and mixtures of emotional experiences. Therefore, the organization of emotional experiences is based on the varying complexity of emotional representations [111,112].

Self-awareness is reinforced by self-care behaviors such as exercise and journal writing. In general, students are an intelligent population and therefore have good study habits. Nevertheless, students with higher levels of EI have a high level of cognitive engagement in many conditions. In this sense, it has been communicated that individuals who show higher levels of EI are likely to be able to identify emotional states in themselves and others. They can use the information to better control the environment according to the situation. In this sense, EI teaches the individual to move from behaviors that seek self-gratification to ones in which gratification is received by understanding the emotional needs of oneself and others [113].

Therefore, the components of EI that regulate stress and emotions in the learning process are [114]:-Self-awareness: This component involves recognizing and understanding one’s own emotions, strengths, weaknesses, and triggers. By being aware of their emotional state, students can identify signs of stress and adopt appropriate measures to regulate them.-Self-regulation: It refers to the ability to manage and control one’s emotions, impulses, and behaviors. In the learning process, self-regulation can help students to cope with stress by regulating negative emotions such as anxiety or frustration. In addition, self-regulation enables students to focus and set in realistic goals, adapting to changing circumstances.-Social awareness: This component refers to being attentive to and understanding the emotions and needs of others. In the context of learning, students with positive social awareness can recognize when peers or teachers are experiencing stress or negative emotions. This recognition enables one to offer support, empathy, or assistance, resulting in a more positive learning environment.-Relationship management: This factor entails effectively managing interpersonal interactions to build positive relationships. In the learning process, students who manage this component positively can establish positive relationships with teachers and peers, increasing collaboration and reducing stress levels through efficient communication and conflict resolution.-Empathy: This is the ability to understand and share feelings with others. Empathic students can recognize and respond appropriately to emotions and stress from teachers and peers. Empathy fosters a supportive and inclusive learning environment.

Altogether, the development of these components of EI in students is instrumental for effectively regulating stress and emotions in the learning process. They can also contribute to a positive and supportive learning environment by empathizing with others and managing relationships effectively. In conclusion, EI components can improve well-being and enhance learning outcomes [114].

### 6.4. The Impact on Talented Individuals

The influence of stress and emotions has to be mentioned on those individuals with special learning qualities. These individuals organize information more accurately and efficiently than others [115,116]. Talented subjects are unique individuals who are endowed with special attributes or characteristics that make them develop, learn and perform skills more easily than others in similar environments [117]. In the fields of art, sports, and science, talented individuals acquire a large number of complex patterns to store new knowledge about what actions or processes should be performed for structured learning [118]. Altogether, due to these qualities, talented individuals can manage stress and emotions very efficiently in the learning process. 

## 7. Discussion

According to the information presented in this review, the impact of stress on the learning process seems to be controversial, impairing or enhancing memory functions [38,39]. The proposed explanation for this discrepancy is that stress could be considered a positive component when it is under control and a negative aspect when control is not possible [1]. Regarding emotions, they present a similar perspective. Emotions related to positive feelings (awareness, understanding, acceptance) favor the learning process; meanwhile, those related to negative feelings (distress, anxiety, threat) impair the learning process [8]. The activation of coping strategies creates key mechanisms for dealing positively with problems and new challenges. Altogether, this response generates positive emotions that are instrumental in the self-regulation of learning.

Therefore, emotions can condition learning, and for this reason, it is important to identify and manage emotions. Regarding the information presented in this review, we propose future research to test some strategies to mitigate the negative impact of stress and emotions in the learning process. Techniques of meditation, mindfulness, and relaxation could be implemented to reduce stress and improve mental health. These techniques could help in cognitive restructuring to identify, evaluate and change wrong thinking. In this context, meditation involves training the mind to focus and redirect thoughts, helping students to reduce stress, improve attention and enhance self-awareness. Meditation sessions, either in the form of guided or silent meditation, can be conducted at the beginning or end of teaching days or integrated into specific activities, such as physical activity or wellness sessions [119]. Mindfulness involves paying attention to the present moment without judgment. This can help students develop emotional regulation, resilience, and empathy. Mindfulness sessions can be integrated into daily routines before the teaching process, such as breathing exercises or guided visualization [120]. Relaxation aims to reduce stress and enhance a state of calmness. Relaxation techniques include breathing exercises, progressive muscle relaxation, or guided imagery. Sessions can be performed individually or in groups and integrated into sport training routines or after study periods. In addition, relaxation techniques could be useful for physical recovery, helping to recharge energy [121]. We propose to test in future research a protocol in which students have to identify the moment of the day when stress appears and perform these techniques routinely. The reduction in anxiety should be verified in this proposal. Finally, deciphering in more detail strategies followed by talented individuals could be an interesting area for future research. In this line, an increase in attention in this study will help to manage academic situations of stress.

Regarding limitations, as a narrative review some articles related to the topic could be missed, although extensive literature has been presented. Since sex can influence feelings of anxiety and frustration, articles regarding this particular aspect were scarce, and this aspect was not afforded in the review. We can speculate that societal expectations, sex stereotypes, and cultural rules can contribute to differential stress experiences in the learning process. In this line, girls may face additional pressure related to academic performance, family and societal expectations, and sex roles. In addition, factors such as sex discrimination can contribute to increasing stress levels among female students [105]. Future research has to address these sex-related factors and study how inclusive learning environments could promote equal opportunities, reducing stress and negative emotions. In addition, other potential limitations include the generalizability of the findings or biases in some of the selected studies. Finally, more physiological research is necessary to correlate anxiety feelings with activation patterns of the CNS. In this context, recent research has revealed new insights into the role of the insular cortex in emotional processing and the interaction between the insula and the amygdala [122]. Changes in these patterns after activation of coping strategies could be an interesting area for future research in neuropsychology and pharmacological interventions.

## 8. Conclusions

Stress and emotions have a main influence on the learning process. The COVID-19 pandemic was the more recent reference to social stress. People isolated in quarantine were at greater risk for anxiety and depression. Students were one of the population segments affected by the learning process. In this context, stress tends to be seen as a negative condition in the learning process because it impairs memory functions. However, other evidence indicates that stress enhances memory encoding. Therefore, the role of stress in encoding information needs further investigation. On the other hand, emotions can influence the learning process. Positive emotions are healthy and can favor optimal learning. Emotional states are regulated by a combined function of nervous, endocrine, and skeletal-muscle systems. In conclusion, modulation of emotions in stress could promote effective learning through focusing attention or solving problems.

## Figures and Tables

**Figure 1 healthcare-11-01787-f001:**
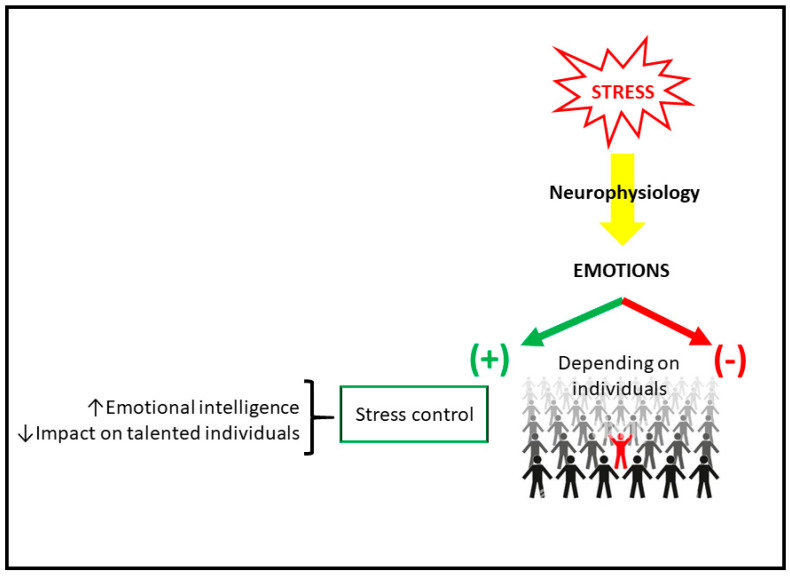
Key points of this review. See text for more details.

**Table 1 healthcare-11-01787-t001:** Emotional aspects. Relationship between functions, components, and organ systems.

Emotional Function	Emotional Component	Physiological Function (Organism Systems)
Event evaluation	Cognitive component	Information processing (CNS)
System regulation	Neurophysiological component	Support (CNS, ANS, NES)
Action preparation and direction	Motivational component (action tendencies)	Executive (CNS)
Communication of reaction and behavioral intention	Motor expression component (facial and vocal expression)	Action (SNS)
Monitoring of internal state and organism-environment interaction	Subjective feeling component (emotional experience)	Monitoring (CNS)

CNS = central nervous system. ANS = autonomic nervous system. NES = neuroendocrine system. SNS = somatic nervous system.

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
