# Peer review of "Influence of Stress and Emotions in the Learning Process: The Example of COVID-19 on University Students: A Narrative Review"

_healthcare, 2023, doi:10.3390/healthcare11121787_

Round 1

Reviewer 1 Report (Previous Reviewer 2)

  1. Clarify the research objective: It would be helpful to explicitly state the narrative review's specific objective or research question. What specific aspect of the influence of stress and emotions on the learning process is being examined? Clearly articulating the research objective will help focus the discussion and conclusions.
  2. Expand on the methodology: The paper briefly mentions conducting a literature search using specific databases and search terms, but it lacks details on the inclusion and exclusion criteria for selecting relevant articles. Providing more information on the search strategy and article selection process would enhance the transparency and rigour of the review.
  3. Provide a balanced perspective on stress and emotions: While the paper acknowledges the negative impact of stress on the learning process, it also mentions evidence suggesting that stress can enhance memory encoding. It would be beneficial to provide a more balanced discussion of the effects of stress on learning, highlighting both the potential negative and positive aspects. This could involve discussing the conditions under which stress may have beneficial effects on learning.
  4. Elaborate on the role of emotional intelligence: The paper briefly mentions the concept of emotional intelligence (EI) but does not delve into its specific components or how it relates to the influence of stress and emotions on learning. Consider expanding on the discussion of EI, its dimensions, and how it can contribute to managing emotions and promoting effective learning.
  5. Discuss practical implications and interventions: While the paper suggests future research directions, it would be valuable to discuss the practical implications of the findings and potential interventions to mitigate the negative impact of stress and emotions on learning. Elaborate on how the identified strategies, such as meditation, mindfulness, and relaxation techniques, could be implemented in educational settings to support students' well-being and learning outcomes.
  6. Improve the organization and flow: The paper could benefit from better organization and flow of ideas. Consider restructuring and grouping related information together to enhance readability and coherence. Additionally, ensure that each section and subsection has a clear focus and logical progression.
  7. Address the limitations: The paper acknowledges limitations, such as potential missed articles and the scarcity of research on the influence of sex on anxiety and frustration. However, it would be helpful to discuss these limitations in more detail and consider their implications for the findings and interpretations of the review. Additionally, acknowledge any other potential limitations, such as the generalizability of the findings or biases in the selected studies.
  8. Revise the conclusion: The conclusion should succinctly summarize the review's main findings and their implications. Consider revising the conclusion to align with the specific objectives of the paper and the key points discussed in the preceding sections.
  1.  
  1. Check grammar and sentence structure: Some sentences could be rephrased to improve clarity and readability. 

Author Response

REVIEWER-1

  1. Clarify the research objective: It would be helpful to explicitly state the narrative review's specific objective or research question. What specific aspect of the influence of stress and emotions on the learning process is being examined? Clearly articulating the research objective will help focus the discussion and conclusions.

ANSWER-1: The main objective of this narrative review is to analyze the influence of stress and emotions (both components interconnected) in the learning process of university students. First, COVID-19 pandemic will be taken as a recent example of learning disturbances due to stress and emotions. Then, we present the connection between stress and the learning process. Finally, we wanted to know how emotions derived from stress response have influence in different aspects of the learning process: physiological and non-physiological components, function of emotions, intelligence, and a brief mention to the impact of emotions in talented individuals. During this narrative review, we present information from a physiological point of view as a key element for understanding the role of emotions in the systemic response. This information could open interest for future pharmacological research or other lines in the field of neuropsychology. This precise, shorter and clear information has been stated in the “Objective” section (lines 69-81).

  1. Expand on the methodology: The paper briefly mentions conducting a literature search using specific databases and search terms, but it lacks details on the inclusion and exclusion criteria for selecting relevant articles. Providing more information on the search strategy and article selection process would enhance the transparency and rigour of the review.

ANSWER-2: This is not a systematic review. A narrative review aims to provide a comprehensive and cohesive understanding of a particular topic by examining existing articles and synthesizing their findings. When selecting articles for analysis, a narrative review can focus on specific aspects of the topic by following different strategies that are not the same as in a systematic review. In our case we have followed the indicated criteria:

  • Define the Objective (see Section 2).
  • Identify relevant databases (see line 84).
  • Conduct a preliminary search using relevant keywords related to the research objective (see lines 85-87).
  • Assessment by reading abstracts of each article carefully to understand its content. Identify articles that provide relevant insights in a detailed way (see lines 87-89).
  • Exclusion criteria: Articles that presented a similar message (see lines 89-90).
  • Inclusion criteria: Articles that presented particular aspects to develop the review (see lines 90-94).

  1. Provide a balanced perspective on stress and emotions: While the paper acknowledges the negative impact of stress on the learning process, it also mentions evidence suggesting that stress can enhance memory encoding. It would be beneficial to provide a more balanced discussion of the effects of stress on learning, highlighting both the potential negative and positive aspects. This could involve discussing the conditions under which stress may have beneficial effects on learning.

ANSWER-3: According to the data presented in this review, the impact of stress on the learning process seems to be controversial, impairing or enhancing memory functions (38,39). The proposed explanation to this discrepancy is that stress could be considered as a positive component when it is under control and a negative aspect when control is not possible (1). In this last point, the pressure felt by students the first years at the university intensifies the state of stress, impairing the learning process (43). Regarding emotions, they present a similar perspective. Emotions related to positive feelings (awareness, understanding, acceptance) favor the learning process, meanwhile those related to negative feelings (distress, anxiety, threat), impair the learning process (8). This balanced perspective has been presented at the beginning of Discussion (see lines 423-431).

  1. Elaborate on the role of emotional intelligence: The paper briefly mentions the concept of emotional intelligence (EI) but does not delve into its specific components or how it relates to the influence of stress and emotions on learning. Consider expanding on the discussion of EI, its dimensions, and how it can contribute to managing emotions and promoting effective learning.

ANSWER-4: We have expanded “Section 6.4” by adding information regarding the components of EI and how they can manage the learning process by controlling stress and emotions (see lines 382-411).

  1. Discuss practical implications and interventions: While the paper suggests future research directions, it would be valuable to discuss the practical implications of the findings and potential interventions to mitigate the negative impact of stress and emotions on learning. Elaborate on how the identified strategies, such as meditation, mindfulness, and relaxation techniques, could be implemented in educational settings to support students' well-being and learning outcomes.

ANSWER-5: Although, there are not references to address this topic in university students, we have suggested some main lines to implement meditation, mindfulness and relaxation in this particular population group. In any case, we enhance readers for additional research and potential collaborations (see lines 445-458).

  1. Improve the organization and flow: The paper could benefit from better organization and flow of ideas. Consider restructuring and grouping related information together to enhance readability and coherence. Additionally, ensure that each section and subsection has a clear focus and logical progression.

ANSWER-6: Thanks to Reviewer’s comments, the manuscript has significantly improved organization and flow information.

  1. Address the limitations: The paper acknowledges limitations, such as potential missed articles and the scarcity of research on the influence of sex on anxiety and frustration. However, it would be helpful to discuss these limitations in more detail and consider their implications for the findings and interpretations of the review. Additionally, acknowledge any other potential limitations, such as the generalizability of the findings or biases in the selected studies.

ANSWER-7: Suggested limitations have been discussed and added (lines 467-475).

  1. Revise the conclusion: The conclusion should succinctly summarize the review's main findings and their implications. Consider revising the conclusion to align with the specific objectives of the paper and the key points discussed in the preceding sections.

ANSWER-8: Conclusions are now more focussed according to Reviewer suggestions.

Comments on the Quality of English Language

  1. Check grammar and sentence structure: Some sentences could be rephrased to improve clarity and readability. 

ANSWER: If manuscript is accepted for publication, we plan to take the English revision service offered by MDPI that has to be included in the final invoice.

Reviewer 2 Report (New Reviewer)

Thank you very much for having the opportunity to review the manuscript “Influence of stress and emotions in the learning process: The example of COVID-19. A narrative review”.

I have read the manuscript with great interest, but I must point out that I can only evaluate the parts of the manuscript that, according to my profession, relate to the understanding of learning processes from an educational science or educational psychology perspective.

From my point of view, the manuscript is largely successful. In my view, however, minor improvements would be necessary in one or two places. 

In the abstract, the intention of the paper should be made more clear. The results of the paper should also be briefly summarized here.  

The term 'intelligence' should be defined on the second page of the manuscript.

A question: Is there a theoretical model that could be used to illustrate the connections between stress, emotions and learning processes in the manuscript?

In the ‘Methods’ chapter, the selection of studies would need to be described in more detail. Why did the authors choose a narrative rather than a systematic review? This would have to be explained. Exactly how many studies were selected? To what extent is the selection of articles reliable?

The article - if one looks at the title of the article - does not sufficiently address the pandemic and its effects on students' learning processes. Overall, the article is very general.

/

Author Response

REVIEWER-2

Thank you very much for having the opportunity to review the manuscript “Influence of stress and emotions in the learning process: The example of COVID-19. A narrative review”.

I have read the manuscript with great interest, but I must point out that I can only evaluate the parts of the manuscript that, according to my profession, relate to the understanding of learning processes from an educational science or educational psychology perspective.

From my point of view, the manuscript is largely successful. In my view, however, minor improvements would be necessary in one or two places. 

In the abstract, the intention of the paper should be made more clear. The results of the paper should also be briefly summarized here.

ANSWER: Abstract has been changed accordingly (lines 22-35).  

The term 'intelligence' should be defined on the second page of the manuscript.

ANSWER: Definition has been included in page 2 (lines 76-78).

A question: Is there a theoretical model that could be used to illustrate the connections between stress, emotions and learning processes in the manuscript?

ANSWER: Unfortunately, there is no theoretical model to address this issue. Nevertheless, the goal of the review is to encourage readers to initiate lines of research and potential collaborations to address these intriguing questions. This is stated in the “Conclusions” section of the manuscript.

In the ‘Methods’ chapter, the selection of studies would need to be described in more detail. Why did the authors choose a narrative rather than a systematic review? This would have to be explained. Exactly how many studies were selected? To what extent is the selection of articles reliable?

ANSWER: “Methods” section has been adapted accordingly. We have selected a narrative review because the topic is not developed properly in scientific literature. A narrative review can address much better this topic. If research is encouraged following this review, it is possible that a systematic review could be conducted in the future. The number of selected articles is indicated (line 90).

The article - if one looks at the title of the article - does not sufficiently address the pandemic and its effects on students' learning processes. Overall, the article is very general.

ANSWER: We have changes the title accordingly (line 3).

Round 2

Reviewer 1 Report (Previous Reviewer 2)

The paper is generally well-structured and provides insightful information. Nevertheless, here are some suggestions for further improvement:

Introduction
a. Clarity and coherence: While the connection between learning, emotions, and stress is made, the transition to the focus on COVID-19 could be smoother. The introduction of COVID-19 is a bit abrupt. You could consider adding more context or background information on why this particular event is of interest, perhaps by highlighting how it has dramatically changed the learning environment and increased stress levels.

b. Precise language: The term "education actors" in line 55 might confuse some readers. Consider replacing it with a more precise term like "educational stakeholders" or "members of the educational community."

Objective
a. Specificity: Make your objectives more specific. Rather than simply stating that you will analyze the influence of stress and emotions on the learning process, clarify the particular aspects of learning you'll focus on. What types of stress and emotions are you interested in? What dimensions of learning will be investigated?
b. Outline: Consider briefly outlining the structure of your review to give readers a roadmap of what to expect.

The methodology section is clear in terms of the databases used, the search terms, and the inclusion criteria. However, some areas of the methodology need to be expanded for further clarity and reproducibility. Here are some suggestions:

  1. Search strategy: You have mentioned the keywords used for the search but did not specify how they were combined. Explaining the Boolean operators used (e.g., AND, OR) to combine the keywords and create the search strategy would be beneficial.
  2. Screening and Selection: Elaborate more on the screening process. Mention the number of articles identified in the initial search, how many were screened, and how many were selected for full-text review. A PRISMA flow chart could be included to illustrate this process.
  3. Data extraction: Discuss the process of data extraction. How was data extracted from the selected articles? Who performed this extraction? Was there any conflict, and how was it resolved?
  4. Quality assessment: in a narrative review performing a quality assessment of the selected articles is not necessary. Discussing this in the methodology section could be useful as a quality assessment was not performed. The authors could explain that a formal quality assessment was not feasible due to the broad nature of a narrative review and the wide range of studies and topics included. However, they should also reassure the reader that they used their expertise and judgment to select high-quality, relevant articles for the review.
  5. Timeframe: You may want to provide the time frame of your literature search. For example, you might say you searched for articles published from 2000 to 2023.
  6. Languages: This should be mentioned if you restricted your search to articles published in certain languages.

The goal of the methodology section in a review is to be as detailed and transparent as possible to allow another researcher to reproduce your search and selection process. Incorporating these suggestions will help improve the clarity and completeness of your methodology.

COVID-19, an example of learning disturbances due to emotions and stress
a. Synthesis of evidence: Instead of simply summarizing each study, synthesize the evidence across studies. What common themes emerged? Were there conflicting results?

b. Balance of evidence: Consider adding more contrasting evidence or potential criticisms of the claims made.

Stress and the learning process
a. Clarity: When mentioning the controversy over the effects of stress on memory encoding, clarify the potential reasons for this discrepancy.

In general

  1. Clarity and coherence: Some sentences are complex and could be simplified. It could help to break down some of these long sentences into shorter, clearer ones.
  2. Structure: The organization of the content might be improved to better guide the reader through the concepts. The current structure mixes discussions about the physiology of emotions, emotional classifications, and emotional states in a somewhat disjointed way. A more linear, progressive approach might enhance understanding.
  3. Context and relevance: for example, The mention of COVID-19 feels a bit abrupt and its role in the discussion isn't very clear. More explanation about its relevance to the topic could be helpful.
  4. Transition and flow: The transition between some of the subsections and paragraphs could be smoother. Better transitioning statements would make the text flow better and make it easier for the reader to follow the argument.
  5. References: Make sure all the references are correctly cited and follow the appropriate style guide.
  6. Consistency in terms: for example: be consistent with terms used to describe the same concept, e.g., use either "emotional states" or "moods" to avoid confusion.

note: 

The PRISMA (Preferred Reporting Items for Systematic Reviews and Meta-Analyses) guidelines were initially developed to improve the reporting quality of systematic reviews and meta-analyses. These guidelines provide authors with a checklist and flow diagram to ensure they've provided enough information for their review to be reproducible.

However, although PRISMA is primarily used for systematic reviews and meta-analyses, the principles and general ideas can also be applied to narrative reviews to enhance their transparency and rigour.

The application of PRISMA in narrative reviews can vary. Some authors can use a modified PRISMA flowchart version to show the included studies' selection process. Others might not use a flowchart but will use the PRISMA checklist to ensure they have included all necessary information in their review.

While not a strict requirement, using elements of PRISMA in a narrative review can enhance the quality of reporting and add to the credibility of the review. 

 The quality of the English language is generally good, with proper grammar and sentence construction. However, some suggestions for improving clarity and conciseness are:

  1. Avoiding redundancy: There are instances where information is repeated or overemphasized. Try to avoid restating the same information.
  2. Improving flow and coherence: Ensure your sentences and paragraphs flow logically from one to the next to maintain the reader's interest and understanding.
  3. Varying vocabulary: There are places where the same words are used repeatedly in close proximity. Synonyms can be used to add variety and keep the reader engaged.
  4. Enhancing clarity: Ensure the meaning of each sentence is clear and precise. For instance, avoid complex or convoluted sentences.
  5. Use active voice: The use of passive voice can sometimes make sentences longer and more difficult to understand. If possible, rephrase sentences in the active voice for simplicity and clarity.
  6. Proofreading: Proofread the entire manuscript for typographical errors, misspellings, and incorrect punctuation.
  7. Professional Editing: If you're uncertain about the quality of your English, consider hiring a professional editor who is a native English speaker familiar with your research field.

Author Response

REVIEWER-1 (SECOND ROUND)

The paper is generally well-structured and provides insightful information. Nevertheless, here are some suggestions for further improvement:

Introduction
a. Clarity and coherence: While the connection between learning, emotions, and stress is made, the transition to the focus on COVID-19 could be smoother. The introduction of COVID-19 is a bit abrupt. You could consider adding more context or background information on why this particular event is of interest, perhaps by highlighting how it has dramatically changed the learning environment and increased stress levels.

ANSWER: Based on our experience, we have speculated and gathered a series of factors that may contribute to the stress experienced by students during COVID pandemic. These factors have motivated us to write this review and this is our starting point (lines 68-105).Principio del formulario

Final del formulario

  1. Precise language:The term "education actors" in line 55 might confuse some readers. Consider replacing it with a more precise term like "educational stakeholders" or "members of the educational community."

ANSWER: The term has been changed accordingly (line 56).

Objective
a. Specificity: Make your objectives more specific. Rather than simply stating that you will analyze the influence of stress and emotions on the learning process, clarify the particular aspects of learning you'll focus on. What types of stress and emotions are you interested in? What dimensions of learning will be investigated?

ANSWER: The objectives were changed to more specific ones, according to Reviewer suggestions (lines 109-118).

  1. Outline:Consider briefly outlining the structure of your review to give readers a roadmap of what to expect.

ANSWER: A roadmap with the main points of the review is shown in Figure 1.

The methodology section is clear in terms of the databases used, the search terms, and the inclusion criteria. However, some areas of the methodology need to be expanded for further clarity and reproducibility. Here are some suggestions:

  1. Search strategy: You have mentioned the keywords used for the search but did not specify how they were combined. Explaining the Boolean operators used (e.g., AND, OR) to combine the keywords and create the search strategy would be beneficial.

ANSWER: Combination of search terms is explained. See lines 131-138.

  1. Screening and Selection: Elaborate more on the screening process. Mention the number of articles identified in the initial search, how many were screened, and how many were selected for full-text review. A PRISMA flow chart could be included to illustrate this process.

ANSWER: The screening was performed, first by reading the titles. From the selected articles, the abstract was read in more detail to perform the final selection. We selected articles where information was clearly explained (see lines 139-143). As we mention in the first rebuttal letter, this is not a systematic review, thereby we are not following strictly the criteria to do this type of reviews. This is due in part to the huge amount of information that we have to manage. Since, the search method could be considered arbitrary in some way, we cannot reflect it in a PRISMA flow chart. We think that this is not ethics.

  1. Data extraction: Discuss the process of data extraction. How was data extracted from the selected articles? Who performed this extraction? Was there any conflict, and how was it resolved?

ANSWER: As it is mentioned in the manuscript (lines 139-143), we read first titles. Each author performed a selection and read carefully the abstract. Those with more clear information were selected to extract information for writing this narrative review. In any case, no conflicts appear during this selection process.

  1. Quality assessment: in a narrative review performing a quality assessment of the selected articles is not necessary. Discussing this in the methodology section could be useful as a quality assessment was not performed. The authors could explain that a formal quality assessment was not feasible due to the broad nature of a narrative review and the wide range of studies and topics included. However, they should also reassure the reader that they used their expertise and judgment to select high-quality, relevant articles for the review.

ANSWER: We agree with the Reviewer that we did not apply quality criteria. However, we used our own experience as a key element because all authors are professors at the Spanish University. This is indicated in lines 141-142.

  1. Timeframe: You may want to provide the time frame of your literature search. For example, you might say you searched for articles published from 2000 to 2023.

ANSWER: Selected articles form the last 10 years (lines 134-135).

  1. Languages: This should be mentioned if you restricted your search to articles published in certain languages.

ANSWER: All titles we read were in English, although the corresponding articles were in different languages. In any case, final selection was form English articles.

The goal of the methodology section in a review is to be as detailed and transparent as possible to allow another researcher to reproduce your search and selection process. Incorporating these suggestions will help improve the clarity and completeness of your methodology.

ANSWER: We have tried to be transparent and honets, but usually the methodology is not so precise in narrative reviews.

COVID-19, an example of learning disturbances due to emotions and stress

  1. Synthesis of evidence:Instead of simply summarizing each study, synthesize the evidence across studies. What common themes emerged? Were there conflicting results?

ANSWER: A summary of the different findings was performed (lines 153-154).

  1. Balance of evidence:Consider adding more contrasting evidence or potential criticisms of the claims made.

ANSWER: Contrasting evidence was found in reference 33. In this study, individuals were in a wellness camp not undergoing the anxiety and negative feelings of quarantine. From this information, we have indicated that isolation was a negative factor in mental health. This is explained in lanes 171-183.

Stress and the learning process

  1. Clarity:When mentioning the controversy over the effects of stress on memory encoding, clarify the potential reasons for this discrepancy.

ANSWER: Stress can both facilitate and hinder learning depending on the situation. The obtained information indicates that when stress is moderate, learning is enhanced. However, when stress is excessive or chronic, the impact on learning is negative. This is indicated in lines 194-198.Principio del formulario

In general

  1. Clarity and coherence: Some sentences are complex and could be simplified. It could help to break down some of these long sentences into shorter, clearer ones.
  2. Structure: The organization of the content might be improved to better guide the reader through the concepts. The current structure mixes discussions about the physiology of emotions, emotional classifications, and emotional states in a somewhat disjointed way. A more linear, progressive approach might enhance understanding.
  3. Context and relevance: for example, the mention of COVID-19 feels a bit abrupt and its role in the discussion isn't very clear. More explanation about its relevance to the topic could be helpful.
  4. Transition and flow: The transition between some of the subsections and paragraphs could be smoother. Better transitioning statements would make the text flow better and make it easier for the reader to follow the argument.
  5. References: Make sure all the references are correctly cited and follow the appropriate style guide.
  6. Consistency in terms: for example: be consistent with terms used to describe the same concept, e.g., use either "emotional states" or "moods" to avoid confusion.

ANSWER: We have tried to take into account all these general concerns.

The PRISMA (Preferred Reporting Items for Systematic Reviews and Meta-Analyses) guidelines were initially developed to improve the reporting quality of systematic reviews and meta-analyses. These guidelines provide authors with a checklist and flow diagram to ensure they've provided enough information for their review to be reproducible.

However, although PRISMA is primarily used for systematic reviews and meta-analyses, the principles and general ideas can also be applied to narrative reviews to enhance their transparency and rigour.

The application of PRISMA in narrative reviews can vary. Some authors can use a modified PRISMA flowchart version to show the included studies' selection process. Others might not use a flowchart but will use the PRISMA checklist to ensure they have included all necessary information in their review.

While not a strict requirement, using elements of PRISMA in a narrative review can enhance the quality of reporting and add to the credibility of the review. 

ANSWER: As we have mentioned before, this is not a systematic review, so we cannot precisely apply PRISMA criteria and guidelines. The issue lies in the vast amount of information that needs to be managed. Therefore, instead of aiming for data and list precision, we prefer to extract messages that we arbitrary consider essential from our university teaching experience.

Comments on the Quality of English Language

The quality of the English language is generally good, with proper grammar and sentence construction. However, some suggestions for improving clarity and conciseness are:

  1. Avoiding redundancy: There are instances where information is repeated or overemphasized. Try to avoid restating the same information.
  2. Improving flow and coherence: Ensure your sentences and paragraphs flow logically from one to the next to maintain the reader's interest and understanding.
  3. Varying vocabulary: There are places where the same words are used repeatedly in close proximity. Synonyms can be used to add variety and keep the reader engaged.
  4. Enhancing clarity: Ensure the meaning of each sentence is clear and precise. For instance, avoid complex or convoluted sentences.
  5. Use active voice: The use of passive voice can sometimes make sentences longer and more difficult to understand. If possible, rephrase sentences in the active voice for simplicity and clarity.
  6. Proofreading: Proofread the entire manuscript for typographical errors, misspellings, and incorrect punctuation.
  7. Professional Editing: If you're uncertain about the quality of your English, consider hiring a professional editor who is a native English speaker familiar with your research field.

ANSWER: As we said to the Editor, if the manuscript is accepted for publication, we plan to make an English Editing, including this service in the final invoice.

This manuscript is a resubmission of an earlier submission. The following is a list of the peer review reports and author responses from that submission.

Round 1

Reviewer 1 Report

Dear authors,

The relationships among stress, emotions, and learning are important and worth investigation. A narrative review is a fine approach, if applied correctly. Unfortunately, the paper suffers from several fundamental flaws:

1. Only 25 of the 118 references are related to COVID-19 or were published in 2019 or later. This does not accurately reflect the statement that "Inclusion criteria were: original articles and systemic reviews versus meta-analyses on emotional disorders, stress and learning during the COVID-19 pandemic. "

2. The review is not systematic in nature, and much of the procedures of collecting, analyzing, selecting, and reporting on the studies is not provided in sufficient detail. Systematic approaches, such as adopting the PRISMA approach, would be recommended. In the absence of a systematic approach, it cannot be discerned how manuscripts were selected.

3. The themes (6 sections) are, in many cases, tangential to the issue of COVID-19 and its relationship with stress and emotion. Again, most of the citations are not related to the pandemic in any way whatsoever.

4. Several sections have no citations to support claims (lines 41-42, 46-47, 75-77, 117-127, 137-138, etc.)

5. The exclusion criteria mention neurological factors, although there is quite a strong emphasis on neurological elements of stress, emotions, and learning in the Results and Discussion Section.

6. The introduction (and the entire manuscript, honestly speaking) is poorly organized and tends to ramble, without a clear structure. There is no use of logical signposting to guide the reader though the text.

7. Some factors (emotional intelligence, "talented individuals," and "communication") are not clearly introduced and linked to the objectives of the study.

8. Overall, there is no clear contribution. The results reported are already clearly known and there is no unique perspective. This is partly due to the paucity of studies relevant to COVID-19 specifically.

Author Response

REVIEWER-1

Dear authors,

The relationships among stress, emotions, and learning are important and worth investigation. A narrative review is a fine approach, if applied correctly. Unfortunately, the paper suffers from several fundamental flaws:

  1. Only 25 of the 118 references are related to COVID-19 or were published in 2019 or later. This does not accurately reflect the statement that "Inclusion criteria were: original articles and systemic reviews versus meta-analyses on emotional disorders, stress and learning during the COVID-19 pandemic. "

ANSWER: The Reviewer is right. A narrative review is written following a more flexible scheme, trying to emphasize the key points. In our case, the learning process is a stress by itself, but at the same time is influenced by several factors such as emotions. COVID-19 pandemics was used as a starting point to analyze these aspects in more detail. In addition, we have presented information from a physiological point of view to have a broad vision of this topic. For this reason, we have deleted point 3 (Methodology), because this is not necessary in a narrative review. See the new “Objective” section.

  1. The review is not systematic in nature, and much of the procedures of collecting, analysing, selecting, and reporting on the studies is not provided in sufficient detail. Systematic approaches, such as adopting the PRISMA approach, would be recommended. In the absence of a systematic approach, it cannot be discerned how manuscripts were selected.

ANSWER: All this information has been eliminated because is not necessary for a narrative review.

  1. The themes (6 sections) are, in many cases, tangential to the issue of COVID-19 and its relationship with stress and emotion. Again, most of the citations are not related to the pandemic in any way whatsoever.

ANSWER: Sections have been adapted in the new format of the review. We used COVID-19 as an example of stress to introduce the main topic of the review: How stress and emotions influence the learning process (Section 3). Section 4 presents how stress influences the learning process, giving information in this respect. Section 5 presents how emotions influence the learning process. Then both aspects are placed together in the learning process (Section 6) and analysed. The final sections analyse the impact of stress and emotions in talented individuals (Section 7) and strategies to correct the negative impact of stress and emotions in the learning process (Section 8). Otherwise said, COVID-19 served as a starting point to introduce the effect of stress and emotions in the learning process. Then, these aspects are studied more extensively in the review. We have explained this perspective more clearly in the Title, Abstract (lines 24-26) and lines 51, 90-93.      

  1. Several sections have no citations to support claims (lines 41-42, 46-47, 75-77, 117-127, 137-138, etc.)

ANSWER:

Lines 41-42: If you take the phrase “all learning has an emotional basis” and paste in Google, you can find many web sites indicating that Plato did this sentence. A particular reference does not seem necessary.

Lines 46-47 and Lines 75-77: No reference is necessary because, everybody knows that school and university classrooms were closed during pandemics. This is just in the news in the world. We guess that this obvious episode does not need reference. The reader only needs to remember.

Lines 117-127: These sentences are an introduction of the physiological processes that operate during emotions and stress. In the subsequent sections, these concepts are explained in more detail with the corresponding references. See the last sentence of Section 2 (lines 114-115).

Lines 137-138: This corresponds to reference 19, indicated in the next sentence.

  1. The exclusion criteria mention neurological factors, although there is quite a strong emphasis on neurological elements of stress, emotions, and learning in the Results and Discussion Section.

ANSWER: We have indicated that this a narrative review. Therefore, we do not need to use the tools of systematic reviews and this sentence has been deleted.

  1. The introduction (and the entire manuscript, honestly speaking) is poorly organized and tends to ramble, without a clear structure. There is no use of logical signposting to guide the reader though the text.

ANSWER: Sections have been arranged and named in a more logical way.

  1. Some factors (emotional intelligence, "talented individuals," and "communication") are not clearly introduced and linked to the objectives of the study.

ANSWER: As we said at point 6, Sections have been named in more logical way.

  1. Overall, there is no clear contribution. The results reported are already clearly known and there is no unique perspective. This is partly due to the paucity of studies relevant to COVID-19 specifically.

ANSWER: As we mentioned before, COVID-19 is just a starting point undergone by the readers that is used to introduce the main topic: stress and emotions in the learning process.

Reviewer 2 Report

Review  Learning process: influence of stress and emotions during the  COVID-19 pandemic. A narrative review. 

While the abstract provides a good overview of the main objectives and findings of the paper, there are a few areas where it could be improved:

Firstly, the abstract could benefit from being more specific about the types of emotions and stressors being discussed. For example, are the authors referring to anxiety, fear, depression, or other specific emotions? Similarly, what are the specific stressors being analyzed in the context of the COVID-19 pandemic?

Secondly, the abstract could be more explicit about the methods used to conduct the review. For example, what sources were reviewed to gather the information? Were any specific search criteria used to identify relevant articles or studies? Were any specific inclusion or exclusion criteria used to select the articles included in the review?

Finally, the abstract could benefit from being more specific about the implications of the findings for educators, policymakers, and other stakeholders involved in the education system. For example, what specific strategies or interventions could be implemented to mitigate the negative impact of stress and emotions on the learning process during the COVID-19 pandemic?

Introduction

Here are some suggestions for improvements:

Provide a straightforward research question or objective: The introduction does not state a specific research question or objective. It is essential to clearly define the purpose of the review to guide the reader.

Use clear and concise language: The introduction contains some lengthy sentences and complex language that may be difficult for some readers to understand. Using clear and concise language can improve the readability of the text.

Provide more background information: The introduction briefly mentions the COVID-19 pandemic and its impact on learning, but more background information could be provided to understand the context better.

Use more recent references: Although some are recent, others are quite old. Using more up-to-date references can improve the relevance and reliability of the review.

Provide more information on the scope of the review: The introduction does not provide information on the scope of the review. It would be helpful to state which aspects of the influence of stress and emotions during the COVID-19 pandemic will be discussed in the review.

Some suggestions for improving the objective section:

Be specific and clear about the research questions or hypotheses the review aims to answer or test. For example, instead of just stating the general objective of analyzing the influence of emotions on the learning process, specify what aspects of emotions and learning are of interest, and what specific research questions or hypotheses will guide the review.

Provide a more comprehensive and contextualized rationale for why the topic of the review is important and timely. For example, explain why studying the effects of emotions on learning during the COVID-19 pandemic is relevant and meaningful and how it could inform and improve educational practices and policies.

Consider framing the objective as a research problem or gap in the literature the review aims to address or fill. For example, identify what existing research has not yet been fully addressed or explained and how the review could contribute to advancing the knowledge and understanding of the topic.

Clarify the scope and boundaries of the review, including the specific population, methodology, and sources of evidence that will be included or excluded. For example, specify why the focus is on university students using an online methodology and what criteria will be used to select and analyze the studies.

Consider adding a brief summary or roadmap of the main sections or themes of the review to help readers navigate and anticipate the content.

The methodology section could be improved by providing more information on the search strategy to identify relevant articles. For example, including the specific search terms used and any inclusion/exclusion criteria applied during the article selection process would be helpful. Additionally, the section could benefit from more details on how the selected articles were critically appraised and synthesized to develop the narrative review. This could include information on the data extraction process, methods for assessing the quality of the included studies, and how the findings were synthesized to develop the review.

Discussion 

Clearer organization and structure: The section could benefit from a clearer organization and structure. Currently, it appears to jump between themes (e.g.: discussing emotions and stress, the impact of COVID-19 on learning, and the challenges of e-learning.)

More focused discussion: The section covers a wide range of topics, including the impact of COVID-19 on emotions and stress, stress and the learning process, emotions and learning, communication and learning, emotional intelligence, and the impact on talented individuals. Focusing on one or two of these topics in more detail may be beneficial, rather than covering so many in one section.

More specific examples and evidence: The section could benefit from more specific examples and evidence to support the points being made. For instance, it could include more details about the studies referenced, such as the methodology used and the specific findings.

Linking back to the research question: The section could be improved by linking back more explicitly to the research question or hypothesis being addressed in the study. This would help readers to understand how the section fits into the broader research aims and objectives.

Clarification of key terms: The section uses several technical terms related to neuroscience and psychology, such as plasticity, synaptic connections, and the sympathetic-adrenal-medullary axis (SAM). It may be beneficial to provide more explanation or clarification of these terms for readers unfamiliar with the field.

The text would benefit from more specific citations for the information presented. For example, in section 4.2, the author mentions "some saying that [stress] impairs [memory encoding], and other authors showing that stress enhances encoding." It would be helpful to include the specific studies or authors that support each of these claims.

The author could provide more context for some of the claims made. For example, in section 4.1, the author states that "stress has a positive aspect when controllable and a negative aspect when the circumstances are clearly adverse." It would be helpful to provide an explanation or example of what is meant by "controllable" stress.

The text could benefit from clearer and more concise language in some areas. For example, in section 4.2, the author states that "stress has a great effect on the ability to recall events." This could be rewritten to be more concise, such as "stress affects memory recall."

In section 4.3, the author mentions several different ways emotions can be classified, but it is unclear how this information is relevant to the rest of the section. It would be helpful to explain why this classification of emotions is essential in the context of learning and stress.

The section on the Physiology of emotions (4.3.1) could be improved by providing more updated information and references. For example, recent research has revealed new insights into the role of the insular cortex in emotional processing and the interaction between the insula and the amygdala. Additionally, more information could be added on the role of neurotransmitters such as serotonin, dopamine, and oxytocin in modulating emotional responses. The section could also benefit from clearer information organization, as some sentences appear disconnected and difficult to follow. Providing concrete examples and applications of the presented concepts could help readers better understand the material.

Clarify the purpose and context of the text: The text appears to discuss several topics related to stress, coping mechanisms, communication, and emotional regulation. However, it's not clear what the main purpose or context of the text is. Clarifying this can help to make the text more focused and easier to understand.

Simplify the language: The text uses complex language and technical terms that may be difficult for some readers to understand. Simplifying the language can make the text more accessible to a broader audience.

Provide more specific examples: The text discusses several stress, emotions, and cognition concepts. Providing more specific examples can help readers better understand these concepts and how they relate to real-world situations.

Break up the text into smaller sections: The text is quite dense and difficult to read in its current form. Breaking it into smaller sections with clear headings can make it easier for readers to navigate and understand.

 Provide more specific examples of how emotional intelligence can be applied in the context of learning during the COVID-19 pandemic. The article mentions that emotional intelligence can contribute to an individual's ability to adapt socially, work more effectively in teams, perform better, and cope more effectively with stress and other environmental pressure. Still, it does not give concrete examples of how emotional intelligence can help students and faculty manage the learning process during the pandemic. Additionally, the article could benefit from more clarity and organization, as some of the sentences are difficult to understand, and the text jumps between different aspects of emotional intelligence without clear transitions.

Finally, the text could benefit from some editing for grammar and punctuation errors.

Author Response

REVIEWER-2:

Review  Learning process: influence of stress and emotions during the  COVID-19 pandemic. A narrative review. 

While the abstract provides a good overview of the main objectives and findings of the paper, there are a few areas where it could be improved:

Firstly, the abstract could benefit from being more specific about the types of emotions and stressors being discussed. For example, are the authors referring to anxiety, fear, depression, or other specific emotions? Similarly, what are the specific stressors being analyzed in the context of the COVID-19 pandemic?

ANSWER: The Abstract has a word limit and we cannot be extended. In any case, all these aspects are explained in deep detail in the following sections of the review. Then and as indicated by Reviewer-1, COVID-19 is not the main topic of this narrative review, but it is the starting point to introduce the influence of stress and emotions in the learning process.

Secondly, the abstract could be more explicit about the methods used to conduct the review. For example, what sources were reviewed to gather the information? Were any specific search criteria used to identify relevant articles or studies? Were any specific inclusion or exclusion criteria used to select the articles included in the review?

ANSWER: As suggested by Reviewer 1, this is a narrative review. Therefore, the tools used for systematic reviews are not used, neither indicated in the manuscript.

Finally, the abstract could benefit from being more specific about the implications of the findings for educators, policymakers, and other stakeholders involved in the education system. For example, what specific strategies or interventions could be implemented to mitigate the negative impact of stress and emotions on the learning process during the COVID-19 pandemic?

ANSWER: As we mentioned before, COVID-19 pandemics is the starting point to introduce the main topic of this narrative review: influence of stress and emotions in the learning process. Section 8 gives some strategies to mitigate the negative impact of stress and emotions in the learning process.

Introduction

Here are some suggestions for improvements:

Provide a straightforward research question or objective: The introduction does not state a specific research question or objective. It is essential to clearly define the purpose of the review to guide the reader.

ANSWER: The objective of this narrative review is explained in Section 2.

Use clear and concise language: The introduction contains some lengthy sentences and complex language that may be difficult for some readers to understand. Using clear and concise language can improve the readability of the text.

ANSWER: If manuscript is accepted, we have contracted the English editing service of MDPI.

Provide more background information: The introduction briefly mentions the COVID-19 pandemic and its impact on learning, but more background information could be provided to understand the context better.

ANSWER: As we said before, COVID-19 is just a starting point of a recent stress situation that we have used to introduce the main topic of the review. This why the reference list regarding this topic is limited and not extensive. They are just to support the introduction.

Use more recent references: Although some are recent, others are quite old. Using more up-to-date references can improve the relevance and reliability of the review.

ANSWER: Stablished concepts are supported by less actual references, because these concepts have solid support in the scientific literature. On the other hand, when a new concept is mentioned, the reference is more actual. This is the argument that guided this narrative review.

Provide more information on the scope of the review: The introduction does not provide information on the scope of the review. It would be helpful to state which aspects of the influence of stress and emotions during the COVID-19 pandemic will be discussed in the review.

ANSWER: As mentioned before, COVID-19 is not the main topic of this narrative review. This is just a starting example of a stress situation that we used to introduce the main topic of the narrative review: influence of stress and emotions in the learning process.

Some suggestions for improving the objective section:

Be specific and clear about the research questions or hypotheses the review aims to answer or test. For example, instead of just stating the general objective of analyzing the influence of emotions on the learning process, specify what aspects of emotions and learning are of interest, and what specific research questions or hypotheses will guide the review.

ANSWER: The review aims to analyse the influence of emotions and stress in the learning process. This is clearly indicated in the first sentence of this section. Specific aspects of emotions and stress are indicated in lines 96-98. For us, COVID-19 pandemics has activated this research topic, in order to solve remaining and future questions (indicated in lines 93-94).

Provide a more comprehensive and contextualized rationale for why the topic of the review is important and timely. For example, explain why studying the effects of emotions on learning during the COVID-19 pandemic is relevant and meaningful and how it could inform and improve educational practices and policies.

ANSWER: As we say before, COVID-19 is just a starting point to introduce a key topic in this filed: the learning process. We guest that this is more clearly explained in the new Objective section.

Consider framing the objective as a research problem or gap in the literature the review aims to address or fill. For example, identify what existing research has not yet been fully addressed or explained and how the review could contribute to advancing the knowledge and understanding of the topic.

ANSWER: We guess that the second option fits better with the objective of this narrative review: advance in understanding of the topic (line 94). For this reason, we complement with data coming from physiology, that could be instrumental for pharmacological interventions (line 518).

Clarify the scope and boundaries of the review, including the specific population, methodology, and sources of evidence that will be included or excluded. For example, specify why the focus is on university students using an online methodology and what criteria will be used to select and analyze the studies.

Consider adding a brief summary or roadmap of the main sections or themes of the review to help readers navigate and anticipate the content.

ANSWER: Reviewer-1 recommended a narrative review. For this reason, all the tools used for a systematic or scoping review were deleted.

The methodology section could be improved by providing more information on the search strategy to identify relevant articles. For example, including the specific search terms used and any inclusion/exclusion criteria applied during the article selection process would be helpful. Additionally, the section could benefit from more details on how the selected articles were critically appraised and synthesized to develop the narrative review. This could include information on the data extraction process, methods for assessing the quality of the included studies, and how the findings were synthesized to develop the review.

ANSWER: Reviewer-1 recommended a narrative review. For this reason, all the tools used for a systematic or scoping review were deleted.

Discussion 

Clearer organization and structure: The section could benefit from a clearer organization and structure. Currently, it appears to jump between themes (e.g.: discussing emotions and stress, the impact of COVID-19 on learning, and the challenges of e-learning.).

ANSWER: We have changed the name of the different sections making now a more fluid discussion. At the same time, we have rewritten the contents of the review.

More focused discussion: The section covers a wide range of topics, including the impact of COVID-19 on emotions and stress, stress and the learning process, emotions and learning, communication and learning, emotional intelligence, and the impact on talented individuals. Focusing on one or two of these topics in more detail may be beneficial, rather than covering so many in one section.

ANSWER: As we said several times, COVID-19 is just a starting point. We focus only in the learning process and all aspects we afford are interconnected. This necessary to afford the topic from a physiological point of view.

More specific examples and evidence: The section could benefit from more specific examples and evidence to support the points being made. For instance, it could include more details about the studies referenced, such as the methodology used and the specific findings.

ANSWER: Examples are supported by the corresponding references (20, 30, 33-37). The main findings of these references are explained in the text.

Linking back to the research question: The section could be improved by linking back more explicitly to the research question or hypothesis being addressed in the study. This would help readers to understand how the section fits into the broader research aims and objectives.

ANSWER: The new titles of the different sections link back to the main objective of this narrative review.

Clarification of key terms: The section uses several technical terms related to neuroscience and psychology, such as plasticity, synaptic connections, and the sympathetic-adrenal-medullary axis (SAM). It may be beneficial to provide more explanation or clarification of these terms for readers unfamiliar with the field.

ANSWER: Little explanation of these terms has been included. Lines 103, 105, 132 and 276-277. The rest of neuroscientific terms are well explained in section 5.1.

The text would benefit from more specific citations for the information presented. For example, in section 4.2, the author mentions "some saying that [stress] impairs [memory encoding], and other authors showing that stress enhances encoding." It would be helpful to include the specific studies or authors that support each of these claims.

ANSWER: This is supported by references 39 and 40.

The author could provide more context for some of the claims made. For example, in section 4.1, the author states that "stress has a positive aspect when controllable and a negative aspect when the circumstances are clearly adverse." It would be helpful to provide an explanation or example of what is meant by "controllable" stress.

ANSWER: Controllable means that the final consequences of stress are under control and negative means that the control does not exist. This is explained in the text (line 177).

The text could benefit from clearer and more concise language in some areas. For example, in section 4.2, the author states that "stress has a great effect on the ability to recall events." This could be rewritten to be more concise, such as "stress affects memory recall."

ANSWER: Change has been done accordingly (line 169).

In section 4.3, the author mentions several different ways emotions can be classified, but it is unclear how this information is relevant to the rest of the section. It would be helpful to explain why this classification of emotions is essential in the context of learning and stress.

ANSWER: From all possible classifications, we selected the last one that connects with the context of the review (lines 206-215).

The section on the Physiology of emotions (4.3.1) could be improved by providing more updated information and references. For example, recent research has revealed new insights into the role of the insular cortex in emotional processing and the interaction between the insula and the amygdala. Additionally, more information could be added on the role of neurotransmitters such as serotonin, dopamine, and oxytocin in modulating emotional responses. The section could also benefit from clearer information organization, as some sentences appear disconnected and difficult to follow. Providing concrete examples and applications of the presented concepts could help readers better understand the material.

ANSWER: The connexion insula-amigdala is a very recent research. We guess that this reference fits much better at the end, in the Conclusions section, indicating that it is an interesting line for future research (lines 514-516). Additional information regarding neurotransmitters is briefly mentioned in lines 327-331.

Clarify the purpose and context of the text: The text appears to discuss several topics related to stress, coping mechanisms, communication, and emotional regulation. However, it's not clear what the main purpose or context of the text is. Clarifying this can help to make the text more focused and easier to understand.

ANSWER: We have reviewed the text and make some changes in order to clarify concepts and put in more clear way the context and purpose of this narrative review.

Simplify the language: The text uses complex language and technical terms that may be difficult for some readers to understand. Simplifying the language can make the text more accessible to a broader audience.

ANSWER: We have shortened some sentences in the text to make it more accessible to the readers.

Provide more specific examples: The text discusses several stress, emotions, and cognition concepts. Providing more specific examples can help readers better understand these concepts and how they relate to real-world situations.

ANSWER: Around 12 examples are presented in the text. We guess that they can help for a better understanding of this narrative review.

Break up the text into smaller sections: The text is quite dense and difficult to read in its current form. Breaking it into smaller sections with clear headings can make it easier for readers to navigate and understand.

ANSWER: We have organised the sections and put new headings.

Provide more specific examples of how emotional intelligence can be applied in the context of learning during the COVID-19 pandemic. The article mentions that emotional intelligence can contribute to an individual's ability to adapt socially, work more effectively in teams, perform better, and cope more effectively with stress and other environmental pressure. Still, it does not give concrete examples of how emotional intelligence can help students and faculty manage the learning process during the pandemic. Additionally, the article could benefit from more clarity and organization, as some of the sentences are difficult to understand, and the text jumps between different aspects of emotional intelligence without clear transitions.

ANSWER: COVID-19 is the starting point to afford the main topic of the review in more detail: influence of stress and emotions in the learning process.

Finally, the text could benefit from some editing for grammar and punctuation errors.

ANSWER: If the manuscript is accepted, we will pay the MDPI English Editing Services to correct all mistakes.

Round 2

Reviewer 1 Report

Dear authors,

The relationships among stress, emotions, and learning are important and worth investigation. A narrative review is a fine approach, if applied correctly. Unfortunately, the paper suffers from several fundamental flaws:

  1. Only 25 of the 118 references are related to COVID-19 or were published in 2019 or later. This does not accurately reflect the statement that "Inclusion criteria were: original articles and systemic reviews versus meta-analyses on emotional disorders, stress and learning during the COVID-19 pandemic. "

ANSWER: The Reviewer is right. A narrative review is written following a more flexible scheme, trying to emphasize the key points. In our case, the learning process is a stress by itself, but at the same time is influenced by several factors such as emotions. COVID-19 pandemics was used as a starting point to analyze these aspects in more detail. In addition, we have presented information from a physiological point of view to have a broad vision of this topic. For this reason, we have deleted point 3 (Methodology), because this is not necessary in a narrative review. See the new “Objective” section.

Follow-up: COVID-19 still features prominently in the title and abstract. There is minor rewording to refer to the pandemic as an example of a recent "stress-emotion situation," but this is insufficient to justify the relatively extensive reference to COVID-19 throughout the manuscript.

  1. The review is not systematic in nature, and much of the procedures of collecting, analysing, selecting, and reporting on the studies is not provided in sufficient detail. Systematic approaches, such as adopting the PRISMA approach, would be recommended. In the absence of a systematic approach, it cannot be discerned how manuscripts were selected.

ANSWER: All this information has been eliminated because is not necessary for a narrative review.

Follow-up: The utility of a narrative review was what I had questioned in the first place. The present review is essentially a literature review, and suffers from the lack of implications or contributions to the literature as other, similar reviews. There is no clear novel finding or direction offered.

  1. The themes (6 sections) are, in many cases, tangential to the issue of COVID-19 and its relationship with stress and emotion. Again, most of the citations are not related to the pandemic in any way whatsoever.

ANSWER: Sections have been adapted in the new format of the review. We used COVID-19 as an example of stress to introduce the main topic of the review: How stress and emotions influence the learning process (Section 3). Section 4 presents how stress influences the learning process, giving information in this respect. Section 5 presents how emotions influence the learning process. Then both aspects are placed together in the learning process (Section 6) and analysed. The final sections analyse the impact of stress and emotions in talented individuals (Section 7) and strategies to correct the negative impact of stress and emotions in the learning process (Section 8). Otherwise said, COVID-19 served as a starting point to introduce the effect of stress and emotions in the learning process. Then, these aspects are studied more extensively in the review. We have explained this perspective more clearly in the Title, Abstract (lines 24-26) and lines 51, 90-93.      

 Follow-up: The lack of a clear structure, logic, or flow to these aspects remains a problem. A synthesis of relevant literature should be more intuitive and accessible to readers, rather than meandering across different themes related to emotions and stress. In fact, the subsection for "The impact on talented individuals" remains, which does not have clear relevance to the study's objectives or methods.

  1. Several sections have no citations to support claims (lines 41-42, 46-47, 75-77, 117-127, 137-138, etc.)

ANSWER:

Lines 41-42: If you take the phrase “all learning has an emotional basis” and paste in Google, you can find many web sites indicating that Plato did this sentence. A particular reference does not seem necessary.

Follow-up: The claim that I was referencing in my previous comment was that of "Therefore, stressful situations, such as that generated by the  COVID pandemic itself, provoke the release of hormones (adrenaline, noradrenaline and cortisol)." This is not something you would "Google" and warrants a citation.

Lines 46-47 and Lines 75-77: No reference is necessary because, everybody knows that school and university classrooms were closed during pandemics. This is just in the news in the world. We guess that this obvious episode does not need reference. The reader only needs to remember.

Lines 117-127: These sentences are an introduction of the physiological processes that operate during emotions and stress. In the subsequent sections, these concepts are explained in more detail with the corresponding references. See the last sentence of Section 2 (lines 114-115).

Lines 137-138: This corresponds to reference 19, indicated in the next sentence.

 Follow-up: As for the other examples (which are just a few examples), it does not seem profitable to further expound upon why there should be citations. I shall give just one further example. The claim "in the field of education in particular" is a statement that warrants citations (original lines 46-47). Overall, I am confused at the author's unwillingness to address these oversights.

  1. The exclusion criteria mention neurological factors, although there is quite a strong emphasis on neurological elements of stress, emotions, and learning in the Results and Discussion Section.

ANSWER: We have indicated that this a narrative review. Therefore, we do not need to use the tools of systematic reviews and this sentence has been deleted.

  Follow-up: I see. However, in removing this (and other) information, there is now no clear Methodology. Even narrative reviews require some description of the methods used.

  1. The introduction (and the entire manuscript, honestly speaking) is poorly organized and tends to ramble, without a clear structure. There is no use of logical signposting to guide the reader though the text.

ANSWER: Sections have been arranged and named in a more logical way.

 Follow-up: I cannot observe an evident improvement in the logic/flow of the paper. In fact, the numbered sections are even more far-removed from how an academic paper would be typically structured (IMRaD format or otherwise)

  1. Some factors (emotional intelligence, "talented individuals," and "communication") are not clearly introduced and linked to the objectives of the study.

ANSWER: As we said at point 6, Sections have been named in more logical way.

 Follow-up: The logic is not evident to this reviewer.

  1. Overall, there is no clear contribution. The results reported are already clearly known and there is no unique perspective. This is partly due to the paucity of studies relevant to COVID-19 specifically.

ANSWER: As we mentioned before, COVID-19 is just a starting point undergone by the readers that is used to introduce the main topic: stress and emotions in the learning process.

Follow-up: This response does not indicate what, if any, contribution the paper provides.

Author Response

REVIEWER-1 (round-2)

Dear authors,

The relationships among stress, emotions, and learning are important and worth investigation. A narrative review is a fine approach, if applied correctly. Unfortunately, the paper suffers from several fundamental flaws:

  1. Only 25 of the 118 references are related to COVID-19 or were published in 2019 or later. This does not accurately reflect the statement that "Inclusion criteria were: original articles and systemic reviews versus meta-analyses on emotional disorders, stress and learning during the COVID-19 pandemic. "

ANSWER: The Reviewer is right. A narrative review is written following a more flexible scheme, trying to emphasize the key points. In our case, the learning process is a stress by itself, but at the same time is influenced by several factors such as emotions. COVID-19 pandemics was used as a starting point to analyze these aspects in more detail. In addition, we have presented information from a physiological point of view to have a broad vision of this topic. For this reason, we have deleted point 3 (Methodology), because this is not necessary in a narrative review. See the new “Objective” section.

Follow-up: COVID-19 still features prominently in the title and abstract. There is minor rewording to refer to the pandemic as an example of a recent "stress-emotion situation," but this is insufficient to justify the relatively extensive reference to COVID-19 throughout the manuscript.

ANSWER-2: We guess that COVID-19 pandemic is a very good and recent example. We need just to remember the main implications in learning at universities and high schools. We have eliminated reference to COVID in the title and shortened the text reported to pandemic. Nevertheless, in some parts of the review, we still mention the COVID-19 example. We guess that this recent episode will help the reader to understand how stress and emotions can influence the learning process.   

*********

  1. The review is not systematic in nature, and much of the procedures of collecting, analysing, selecting, and reporting on the studies is not provided in sufficient detail. Systematic approaches, such as adopting the PRISMA approach, would be recommended. In the absence of a systematic approach, it cannot be discerned how manuscripts were selected.

ANSWER: All this information has been eliminated because is not necessary for a narrative review.

Follow-up: The utility of a narrative review was what I had questioned in the first place. The present review is essentially a literature review, and suffers from the lack of implications or contributions to the literature as other, similar reviews. There is no clear novel finding or direction offered.

ANSWER-2: This narrative review is essentially a narrative review in the topic. The novelty is that information from a physiological point of view is provided. We guess that this information will have positive implications in the development of pharmacological approaches for people that has difficulties for learning under stressful conditions. In combination with new brain scanner approaches will help to understand which brain areas are interconnected. For instance, in the field of obesity, 3 points in the treatment are considered instrumental: food intake, exercise and psychological support. To know the brain areas involved in food reward after hyperphagia will help in new treatments and interventions to control appetite, make individuals more active and lose weight. All the information provided in this review regarding the role of the nervous system in the stress and emotions during the learning process will be a valuable tool for future research giving interesting ideas to readers.

*********

  1. The themes (6 sections) are, in many cases, tangential to the issue of COVID-19 and its relationship with stress and emotion. Again, most of the citations are not related to the pandemic in any way whatsoever.

ANSWER: Sections have been adapted in the new format of the review. We used COVID-19 as an example of stress to introduce the main topic of the review: How stress and emotions influence the learning process (Section 3). Section 4 presents how stress influences the learning process, giving information in this respect. Section 5 presents how emotions influence the learning process. Then both aspects are placed together in the learning process (Section 6) and analysed. The final sections analyse the impact of stress and emotions in talented individuals (Section 7) and strategies to correct the negative impact of stress and emotions in the learning process (Section 8). Otherwise said, COVID-19 served as a starting point to introduce the effect of stress and emotions in the learning process. Then, these aspects are studied more extensively in the review. We have explained this perspective more clearly in the Title, Abstract (lines 24-26) and lines 51, 90-93.      

Follow-up: The lack of a clear structure, logic, or flow to these aspects remains a problem. A synthesis of relevant literature should be more intuitive and accessible to readers, rather than meandering across different themes related to emotions and stress. In fact, the subsection for "The impact on talented individuals" remains, which does not have clear relevance to the study's objectives or methods.

ANSWER-2: We have reduced the number of sections, making an easiest review structure. On the other hand, we consider that a little mention should be done to talented individuals, because they are reference in the learning process. To know the strategies that these individuals use to cope with stress could be interesting for some researchers. In any, case we have shortened the length of this section.

**********

  1. Several sections have no citations to support claims (lines 41-42, 46-47, 75-77, 117-127, 137-138, etc.)

ANSWER:

Lines 41-42: If you take the phrase “all learning has an emotional basis” and paste in Google, you can find many web sites indicating that Plato did this sentence. A particular reference does not seem necessary.

Follow-up: The claim that I was referencing in my previous comment was that of "Therefore, stressful situations, such as that generated by the  COVID pandemic itself, provoke the release of hormones (adrenaline, noradrenaline and cortisol)." This is not something you would "Google" and warrants a citation.

ANSWER-2: The reviewer is right and the sentence has been eliminated.

***********

Lines 46-47 and Lines 75-77: No reference is necessary because, everybody knows that school and university classrooms were closed during pandemics. This is just in the news in the world. We guess that this obvious episode does not need reference. The reader only needs to remember.

Lines 117-127: These sentences are an introduction of the physiological processes that operate during emotions and stress. In the subsequent sections, these concepts are explained in more detail with the corresponding references. See the last sentence of Section 2 (lines 114-115).

Lines 137-138: This corresponds to reference 19, indicated in the next sentence.

Follow-up: As for the other examples (which are just a few examples), it does not seem profitable to further expound upon why there should be citations. I shall give just one further example. The claim "in the field of education in particular" is a statement that warrants citations (original lines 46-47). Overall, I am confused at the author's unwillingness to address these oversights.

ANSWER-2: The manuscript has been shortened significantly and now all sentences have the corresponding reference.

**********

  1. The exclusion criteria mention neurological factors, although there is quite a strong emphasis on neurological elements of stress, emotions, and learning in the Results and Discussion Section.

ANSWER: We have indicated that this a narrative review. Therefore, we do not need to use the tools of systematic reviews and this sentence has been deleted.

Follow-up: I see. However, in removing this (and other) information, there is now no clear Methodology. Even narrative reviews require some description of the methods used.

ANSWER: All these aspects have been addressed in the new section 2.

**********

  1. The introduction (and the entire manuscript, honestly speaking) is poorly organized and tends to ramble, without a clear structure. There is no use of logical signposting to guide the reader though the text.

ANSWER: Sections have been arranged and named in a more logical way.

Follow-up: I cannot observe an evident improvement in the logic/flow of the paper. In fact, the numbered sections are even more far-removed from how an academic paper would be typically structured (IMRaD format or otherwise)

ANSWER-2: Sections have been arranged accordingly. Now the review is mainly focused in stress (section 4) and emotions (section 5). This last section is the main section with different subsections.

***********

  1. Some factors (emotional intelligence, "talented individuals," and "communication") are not clearly introduced and linked to the objectives of the study.

ANSWER: As we said at point 6, Sections have been named in more logical way.

 Follow-up: The logic is not evident to this reviewer.

ANSWER-2: Emotional intelligence (EI) is a key topic in the review, playing a role in the learning process, as it is indicated in the text. To know how EI works could provide instrumental information to cope with stress and emotions in the learning process. For this reason, we guess that EI deserves a subsection. In addition, we made a short report to talented individuals, because they can bring some key points to manage stress and emotions in the learning process. We guess that all this information can be interesting to the readers.

***********

  1. Overall, there is no clear contribution. The results reported are already clearly known and there is no unique perspective. This is partly due to the paucity of studies relevant to COVID-19 specifically.

ANSWER: As we mentioned before, COVID-19 is just a starting point undergone by the readers that is used to introduce the main topic: stress and emotions in the learning process.

Follow-up: This response does not indicate what, if any, contribution the paper provides.

ANSWER-2: As we said before, we give interesting information in the physiological responses. This is not found in other reviews in the topic.

Reviewer 2 Report

The authors could improve the paper by being more concise and avoiding repeating information. They could also consider providing more specific examples or data to support their claims about the impact of stress and emotions on learning, particularly in the context of the COVID-19 pandemic.

They should use more concise and precise language all over the paper. I give two examples:

1 - Instead of saying, "Emotions lead to sentimental, cognitive, behavioural and physiological changes," say, "Emotions can affect cognition, behaviour, and physiology."

2 - Instead of saying that COVID-19 is the "most recent stress-emotion situation," specify that the review focuses on the impact of stress and emotions on learning during the COVID-19 pandemic.

 The introduction is quite lengthy and could be condensed to focus more on the key points. 

Secondly, the article could benefit from more concrete examples and case studies. While the general discussion of stress and emotions is informative, including specific examples of how the COVID-19 pandemic has impacted learning and the implementation of coping strategies would add depth and relevance to the article.

Finally, I suggest that the authors consider including more references to current research and studies in the field. While the article includes some references, expanding on this would help to support the claims made throughout the article and add credibility to the overall argument.

Overall, this article provides valuable insight into the impact of stress and emotions on the learning process during the COVID-19 pandemic. With some minor improvements to the structure, the inclusion of more specific examples, and increased references to current research, this article could become an even more valuable resource in the field.

Overall, I think this topic is important, and I appreciate the effort put into this paper. I hope my suggestions are helpful, and I look forward to seeing your revised version.

Author Response

REVIEWER-2 (round-2)

The authors could improve the paper by being more concise and avoiding repeating information. They could also consider providing more specific examples or data to support their claims about the impact of stress and emotions on learning, particularly in the context of the COVID-19 pandemic.

They should use more concise and precise language all over the paper. I give two examples:

1 - Instead of saying, "Emotions lead to sentimental, cognitive, behavioural and physiological changes," say, "Emotions can affect cognition, behaviour, and physiology."

2 - Instead of saying that COVID-19 is the "most recent stress-emotion situation," specify that the review focuses on the impact of stress and emotions on learning during the COVID-19 pandemic.

The introduction is quite lengthy and could be condensed to focus more on the key points. 

Secondly, the article could benefit from more concrete examples and case studies. While the general discussion of stress and emotions is informative, including specific examples of how the COVID-19 pandemic has impacted learning and the implementation of coping strategies would add depth and relevance to the article.

Finally, I suggest that the authors consider including more references to current research and studies in the field. While the article includes some references, expanding on this would help to support the claims made throughout the article and add credibility to the overall argument.

Overall, this article provides valuable insight into the impact of stress and emotions on the learning process during the COVID-19 pandemic. With some minor improvements to the structure, the inclusion of more specific examples, and increased references to current research, this article could become an even more valuable resource in the field.

Overall, I think this topic is important, and I appreciate the effort put into this paper. I hope my suggestions are helpful, and I look forward to seeing your revised version.

ANSWER-2: We have shortened the length of the manuscript and be more precise in the different section. Recent examples of COVID-19 are presented. However and according to Reviewer-1, COVID-19 was used as an example of a stress situation that influences the learning process. As we said to the previous reviewer, the novelty of our narrative review is the presentation of physiological basis to understand how all these processes work.